



# Sensitivity of WRF-Chem model resolution in simulating particulate matter in South-East Asia

Adedayo Rasak Adedeji[1], Lalit Dagar[2], Mohammad Iskandar Petra[1], Liyanage C.De Silva[1], Zhining Tao[3,4]

[1]Faculty of Integrated Technologies, Universiti Brunei Darussalam, Jalan Tungku Link, BE1410, Brunei Darussalam

[2]Environmental Studies, Faculty of Arts and Social Sciences, Universiti Brunei Darussalam, Jalan Tungku Link, BE1410, Brunei Darussalam

[3]Universities Space Research Association, Columbia, MD, USA

[4]NASA Goddard Space Flight Center, Greenbelt, MD, USA

*Correspondence to*: Adedayo Rasak Adedeji (rasaqdayo@gmail.com)

**Abstract.** Frequent haze episodes commonly caused by biomass burning has been a concern in South East Asia. One of such events was the June 2013 severe haze in the region. This study assessed the ability of WRF-Chem in capturing the spatial variability and concentrations of particulate emissions during this period. It analyzed the regional biomass burning emissions and its transport leading to higher particulate matter levels in the region. In order to analyze the effect of grid scale, the horizontal resolution of the simulation was varied between low-resolution (100 km) and high-resolution (20 km). Evaluations

of the simulations were made against meteorological observations pertinent to emission and transport of particulate matter, including surface and vertical air profile variables such as temperature, relative humidity, and wind speed and direction. Particulate matter (PM10 and PM2.5) levels were evaluated using ground measurements in Brunei and Singapore respectively. The meteorological parameters were adequately represented across the model simulations. Increasing the horizontal resolution of the simulations generally improved the representation of meteorology and air quality  but some progronostic variables

maintained similar or better performance with coarse resolution simulation. With the high-resolution simulation,  PM10 concentration in Brunei had a correlation coefficient around 0.4, and the simulated PM2.5 level in Singapore had correlation coefficient around 0.9. Whereas, the low-resolution simulation had correlation coefficients around  0.2 and 0.8 for PM10 and PM2.5 levels at Brunei and Singapore, respectively. Both simulations could not repeat aerosol optical depth (AOD) from reanalysis unless the biomass burning emissions were enhanced. An enhacement factor of 6 with high resolution simulation

gave PM10 and PM2.5 correlations around 0.6 and 0.9 in Brunei and Singapore respectively.





## 1 Introduction

Biomass burning is very predominant in the South-East Asian region due to agricultural activities including frequent slash and clearing of the residual crop in preparation for the new planting season (Oozeer et al., 2016). These activities lead to higher emission of different air pollutants and particulate matter across the region. Regional biomass burning could be observed using remote sensing such as two-dimensional images and thermal anomalies captured by satellites. The June 2013 haze in South East Asia was evident from satellite imageries (NASA, 2013). During this period, air quality data from the Department of

Environment, Parks and Recreation, Brunei Darussalam (JASTRe) and National Environment Agency, Singapore (NEA), revealed that Pollution Standard Index (PSI) records exceeded the unhealthy levels and the WHO (World Health Organization) guidelines on several occasions in Brunei and Singapore, respectively (Department of Environment Parks and Recreation (JASTRe), 2013; National Environment Agency, 2013).

Previous studies of the June 2013 haze episode mostly included statistical studies of the effect of particulate matter on visibility

and air quality (Betha et al., 2014; Bhardwaj et al., 2016; Dotse et al., 2016, 2018; Gaveau et al., 2015; Lee et al., 2017; Lin et al., 2014). Oozeer et al. (2016) numerically studied and analyzed convective mechanisms responsible for the uplift and transport of the particulate emissions from Sumatra over to Peninsular Malaysia during the 2013 event (Oozeer et al., 2016). The correlation coefficient (r) recorded by Oozeer et al. (2016) was 0.504 at Johor using Morrison (2 moment) microphysics ((Morrison et al., 2009) and Mellor-Yamada-Janjić (MYJ) boundary layer scheme in their simulation. The other stations

reported lower correlation coefficients than 0.504.

Recently in the Model Inter-Comparison Study for Asia (MICS-Asia) Phase III, Gao and co-workers noted that most models used to study air quality in Asia were developed in other locations such as Europe and USA. Therefore, the parametizations and assumptions in these models may be suited only for those locaions other than Asia. This emphasizes the need for rigorous investigations on the performance, sensitivities and uncertainties of these models in Asia in order to improve them for local

applications (Gao et al., 2018). Following this objective, our study analyzed the effect of grid resolution in simulating the particulate matter during the haze episode in the South-East Asian region using the Weather Research and Forecasting model with Chemistry (WRF-Chem) (Grell et al., 2005). Among the biomass burning emissions, particular attention was given to suspended particulate matter concentrations (less than 2.5 and 10 microns in size - PM2.5 and PM10) since their levels usually exceeded other emissions (Radojevic and Hassan, 1999).

While emission inventory resolution has an impact in resolving air quality in different regions, Tan et al emphasized that meteorological variables have significant effect on the transport, diffusion, mixing and deposition processes of pollutants such as particulate matter (Tan et al., 2014). Also, X Tie et al. reported that, including emission resolution, the model resolution impacts pollutant simulation results largely due to the variability in the resultant transport processes and meteorological



conditions. Thus, aside emissions, the representation of meteorological variables such as temperature, relative humidity and wind speed and direction at different grid scales influences particle matter distribution and transport (Tie et al., 2010).

This study also assessed the effect of increasing fire emissions in our simulation on particulate matter distribution by modifying the biomass burning enhancement factor in the PREP-CHEM-SRC which was set to 1.3 as default.

In this work, we aim:

1. To simulate the formation, deep convection and long-range transport of the biomass burning emissions that resulted in higher particulate matter levels over the South-East Asia region.

2. To identify the meteorology that caused and intensified the transport of biomass burning emissions from Sumatra and parts of Borneo to other regions in South-East Asia during the haze episode.

3. To analyze the response of meteorology and particulate matter simulation to horizontal grid resolutions.

4. To assess spatial distribution and representation of particulate matter levels across South-East Asia during the severe haze event using a modified biomass burning enhancement factor in the model.

This paper was organized as follows: section 2 described the WRF-Chem model configurations. Section 3 presented the data used to run the model such as initial boundary data and emission input, and evaluation datasets which were mostly obtained from radiosonde and ground observations. Section 4 detailed the analysis and comparison of results, while section 5 gave a summary and conclusion on the study.

## 2 Weather Research and Forecasting Model with Chemistry (WRF-Chem)

### 2.1 Model Configuration

The Weather Research and Forecast (WRF) model coupled with Chemistry (WRF-Chem model v 3.9) (Grell et al., 2005) was used to simulate the biomass burning emissions over the South East Asian region in June 2013. Simulations were carried out using WRF-Chem over a single domain covering Sumatra, Java, Borneo and Malaysian Peninsula with a horizontal grid resolution of 100 km x 100 km (40 x 40 grids) for the lower resolution simulation and 20 km x 20 km (200 x 200 grids) for higher resolution simulations. The vertical level configuration was adapted from (Lin et al., 2009) and consisted of 50 sigma levels. The lowest level was set at about 16 m above the surface while the model top was set at 50 hPa. We maintained the same vertical resolution for all experiments.


## 2.2 Model Chemistry and Physics

In all simulations, the 2-moment Morrison scheme (list references) was used to represent the resolved-scale microphysics. The radiative processes (both longwave and shortwave) were represented by the Rapid Radiative Transfer Model for general circulation models (GCMs) longwave and shortwave radiation schemes (RRTMG) (Mlawer et al., 1997; Pincus et al., 2003) respectively. These two schemes can interact with WRF-Chem to include effects of aerosols on radiation through absorbing and scattering. Convective parameterization was performed by the Grell 3D cumulus scheme, which was an improved version of the Grell-Devenyi (GD) cumulus scheme (Grell and Dévényi, 2002). In addition, the Grell 3D cumulus scheme allows for feedback from parameterized convection to the radiation schemes. Mellor-Yamada-Janjić (MYJ) boundary layer scheme (Hong and Pan, 1996) was adopted, and the moisture and thermal fluxes from the surface was obtained using the Noah land-surface model (Chen and Dudhia, 2001). We selected the Regional Acid Deposition Model, 2nd generation (RADM2) (Stockwell et al., 1990) gas-phase chemistry mechanism coupled with the Modal Aerosol Dynamics Model for Europe (MADE), which included the Secondary Organic Aerosol Model (SORGAM) (Schell et al., 2001), for the simulation.

The MADE/SORGAM module is a modal scheme that describes three log-normally distributed modes to simulate particle size distribution: the Aitken mode (<0.1 μm diameter), the accumulation mode (0.1-2 μm diameter), and the coarse mode (>2 μm diameter), and predicts mass and number concentrations for each aerosol mode. In each mode, particles are assumed to have the same chemical composition (internally mixed), while they are externally mixed among different modes (Zhao et al., 2010). The aerosol direct effect was turned on by activating aerosol radiative feedback in the simulation.

## 3 Meteorological and Emission Data

### 3.1 Data for Model Set-up and Emission Input

This study used meteorological initial and boundary condition from the National Center for Environmental Prediction FiNal reanalysis (NCEP-FNL) for the initialization of meteorological variables (NOAA/NCEP, 2000). The NCEP-FNL has a horizontal resolution of 1-degree by 1-degree prepared operationally every six hours. The dataset is available on the surface and at 26 mandatory (and other pressure) levels from 1000 hPa to 10 hPa. Parameters include surface pressure, sea level pressure, geopotential height, temperature, sea surface temperature, soil values, ice cover, relative humidity, u- and v- winds, vertical motion, vorticity, and ozone (NOAA/NCEP, 2000).

The trace gas and aerosol emissions fields were preprocessed using an adapted version of PREP-CHEM-SRC (Freitas et al., 2011). PREP-CHEM-SRC integrates emissions inventories and forest fire hotspots onto a specified domain. WRF-Chem needs three emissions input inventories to simulate the air pollution: (1) anthropogenic emissions; (2) biomass burning emissions and; (3) biogenic emissions. The biomass burning emissions were estimated using the Brazilian Biomass Burning Emissions



Model (3BEM); (Longo et al., 2010), which were injected into the atmosphere using the plume rise model described by Freitas and co-workers ((Freitas et al., 2006, 2007, 2010). Daily fires detected from different satellites could be utilized and processed by the 3BEM. The forest fire hotspots database integrated into the model typically includes a combination of the MODIS C6 fire location database and the Wildfire Automated Biomass Burning Algorithm (WFABBA) database (University of Wisconsin-Madison, 2013). Aqua and Terra products from MODIS with 1000m resolution (at nadir) have been adopted in many cases but small fires that are common in the Borneo and South-East Asia region are less likely to be captured in this product. We have leveraged on the combination of similar product with higher resolution – Visible Infrared Imaging Radiometer Suite (VIIRS) carried on the Suomi-NPP satellite with a 375m resolution (at nadir), to augment and enhance the fire detection database fed into the 3BEM. The 3BEM filters all the fire products and removes fires within a kilometer of each other to avoid double counting between fire datasets. The anthropogenic emissions were obtained from two global inventory datasets – REanalysis of TROpospheric chemical composition over the past 40 years (RETRO) and Emissions Database for Global Atmospheric Research (EDGAR).

The biogenic emissions data is provided by the Model of Emissions of Gases and Aerosols from Nature version 2.1 (MEGAN2.1). MEGAN2.1 is a model framework for a resolution of 1 km. This model estimates natural emissions that occur in a terrestrial ecosystem (Guenther et al., 2012).

Chemical initial and boundary conditions are needed to account for initial concentrations and inflow/background concentrations. WRF-Chem uses idealized chemical profile generated from the NALROM simulation (based upon northern hemispheric, mid-latitude, clean environment conditions). In this study, the initial chemical boundary condition was set in WRF-Chem using '**mozbc**' tool (MOZART GCM output, reference).

To ensure that the model did not violate the Courant- Friedrichs-Levy (CFL) stability criterion (Courant et al., 1928) the time step of the simulation was set at 600 and 120 seconds for the low-resolution (100km) and high-resolution (20km) simulations respectively – following the recommended maximum of $6 \times$ grid spacing. The WRF-Chem simulations were initialized at 00UTC on 15 June 2013 and ended at 00UTC on 30 June 2013. The emission inventories were updated for each day of simulation and the model outputs were generated every hour. The simulations were restarted at the beginning of each day with initial values from the previous run.

### 3.2 Air Quality and Meteorological Data for Validation

Chemical transport models such as WRF-Chem serve as a tool to quantify and establish the deterministic relationship between the amount of emissions and measured concentrations. The performance of the model must be measured and quantified in order to establish the confidence of such relationship (Kumar et al., 2015). Thus, ground measurements were used to evaluate the model performance of the simulations in this study. Ground observations for PM10 were obtained from Department of Environment and Parks (JASTRe) for 4 locations (Bandar Seri Begawan, Tutong, Kuala Belait and Temburong) in Brunei





Darussalam (Department of Environment Parks and Recreation (JASTRe), 2013). PM2.5 data were extracted from the historical readings of the Singapore online database for the 4 available locations (North, East, West, and Central) (National Environment Agency, 2013). The PM2.5 values were reported at 3-hrs interval while the frequency increased to hourly during

peak periods of haze. The daily averages of PM10 (Brunei) and PM2.5  (Singapore) were used for the model evaluations.

Radiosonde data provided by the Department of Atmospheric Science at the University of Wyoming were obtained for Brunei Airport weather station (4.93° N, 114.93° E) and Singapore station (1.36º N, 103.98º E) to evaluate the WRF-Chem simulated atmospheric temperatures (ºC), relative humidity (%), wind speed (m/s) and wind direction (degree) (University of Wyoming, 2013). Based on the geographical location of Singapore and Brunei Darussalam in South East Asia, evaluating the simulated

convective mechanism in both region is very suitable in the assessment of the June 2013 haze event in the region. Data availability can be seen in Table 1.

## 4 Evaluation of the Simulated Model for the June 2013 Haze Episode

### 4.1 Grid Staggering and Meteorological Evaluation

Aside considering model resolutions, performance of model simulation is influenced by the arrangements of prognostic variables on staggered grids relative to observation. It is also influenced by the interpolation method (e.g. nearest neighbor, bilinear interpolation) adopted in calculating variables from simulations.

In WRF and WRF-Chem model, grid cells are in rectangular distribution and according to Arakawa and Lamb C grid staggering (see Figure 1 – top left) introduced by Arakawa and Lamb (Arakawa and Lamb, 1977). The height (h) is evaluated

at the intersection of grid cells while components of wind variables (u, v) are evaluated at mid-points between grid cells. Scalar variables such as thermodynamic variables (e.g. pressure and temperature) are evaluated at the theta points - not staggered and at the grid center (Collins et al., 2013).

Assume a series of grid cells (A, B and C) with different horizontal resolutions; A < B < C in the ratio 1:4:3 (see Figure 1 – top and bottom left). Consider the relative position of a station at theta point (grid center and position of static variables) of

cell C to theta points of cell B and A in case A (Figure 1 – top left) and B (Figure 1 – bottom left).

In case A, using nearest neighbor technique, a model simulation with coarse resolution A would evaluate static variables in cell C as theta point at the grid center of A. Also, a model simulation with B resolution would evaluate static variables at the station in cell C as theta point in cell B. On the other hand with B resolution, bilinear tecnhnique would evaluate the theta point in the four grid cells based on proximity to the station in cell C and by interpolation. In this case, bilinear interpolation method



with B resolution would have better evaluation of static variables in cell C than nearest neighbor technique. Lower resolution in cell A using nearest neighbor would have the worst performance.

In case B, the nearest neighbor technique would yield same result as case A with B resolution. Whereas, result would be different for bilinear interpolation method which would follow similar procedure in case A by calculating static variables at the new station in another cell C by interpolating values at closest four grid cells. While result in the coarse resolution A using

nearest neighbor would also remain same as case A, it is noteworthy that the result would outperform nearest neighbor method using finer resolution B. Brunei station (see Figure 1 – bottom left) has similar characteristics as case B. Thus, some surface prognostic variables having better results in lower resolution than higher resolution might be attributed to this effect.

Ground measurements at the Brunei Airport station were evaluated against surface parameters from simulations such as the temperature at 2 m, surface relative humidity, wind speed and direction at 10 m. Also, the vertical air profile from the

simulations were evaluated against the sounding (radiosonde) data for Singapore and Brunei Airport provided by the Department of Atmospheric Science at the University of Wyoming (University of Wyoming, 2013). Vertical profiles of temperature (°C), relative humidity (%), wind speed (m/s and wind direction (degree) were evaluated. The wind speed (m/s), wind direction (degree), relative humidity (%), atmospheric temperature (°C), pressure (hPa) and the geopotential height (m) corresponding to Singapore station (1.36º N, 103.98º E) and Brunei Airport weather station (4.93° N, 114.93° E) were extracted

between 15th to 30th June 2013, and compared to the sounding data. The statistical metrics used to evaluate the model performance are the Pearson correlation coefficient *(r)* (Eq. 1), root mean square error (RMSE) (Eq. 2), normalized root mean square error (NRMSE) (Eq. 3) and normalized mean bias factor (NMBF) (Eqs. 4a. and 4b.). The metrics are defined as follows, with $N$ being the number of model and observation pairs, $M$ the model and $O$ the observation values, and $\sigma_M$ and $\sigma_O$ the standard deviations of modeled and observation values, respectively:


$$r = \frac{1}{N-1} \sum_{i=1}^{N} \left(\frac{M_i - \bar{M}}{\sigma_M}\right)\left(\frac{O_i - \bar{O}}{\sigma_O}\right) (1)$$

$$RMSE = \sqrt{\frac{\sum_{i=1}^{N}(M_i - O_i)^2}{N}} (2)$$

$$NRMSE = \frac{RMSE}{O_{max} - O_{min}} (3)$$

$$\bar{M} > \bar{O}; NMBF = \frac{\bar{M}}{\bar{O}} - 1 (4a)$$

$$\bar{M} < \bar{O}; NMBF = 1 - \frac{\bar{O}}{\bar{M}} (4b)$$




Tables 2, 3, and 4  show the mean values of the correlation coefficients ($r$), root mean square error (RMSE), normalized root mean square error (NRMSE) and normalized mean bias factor (NMBF) obtained from the 15-day analysis of observation and WRF-Chem simulated data - 100 km and 20 km resolution simulations with nearest neighbor and bilinear interpolation estimations. The tables in the supplement show daily evaluation for meteorology at Brunei and Singapore stations (see S1-S8)

### 4.1.1 Temperature


The model was evaluated by comparing the surface temperature (the temperature at 2 m in Celsius) from the WRF-Chem simulations to the observations at Brunei International Airport (BIA) provided by the Brunei Darussalam Meteorological Department (BDMD). Figure 2 shows the time series comparison of temperature at 2 m (°C) of the observed hourly temperature to the WRF-Chem simulations at Brunei station. In Brunei station, surface temperature simulated by both high and low horizontal resolution simulation had very good correlation (above 0.9) with the observation. However, it is noticeable from the comparison of time-avergaed surface temperature distrubtion across the region between reanalysis and simulations that higher resolution simulations resolved terrain with temperature gradient better at regions in Sumatra and Borneo with higher altitudes (Figure 3). Although WRF-Chem_20km had lower error and bias in Brunei station using nearest neighbor calculation method with RMSE, NRMSE and NMBF of 1.760 °C, 0.154 and -0.039 respectively, WRF-Chem_100km simulation had slightly better correlation (0.924 against 0.908). Also, the bilinear interpolation calculation yielded better results in WRF-Chem_100km than WRF-Chem_20km for surface temperature in Brunei station (see Table 2).



Considering air temperature in Brunei, both low and high-resolution simulations maintained similar and very good results (correlation coefficient of 0.999) with barely any difference between nearest neighbor and bilinear interpolation estimation methods. Overall, the bilinear interpolation in the high-resolution simulation gave the least error (RMSE and NRMSE of 1.759 °C and 0.016 respectively), while nearest neighbor estimation gave the least bias value of -0.011 (Table 3).


In Singapore, the air temperature simulation results were similar to Brunei. Both low and high-resolution simulations had the same correlation coefficient of 0.999 with nearest neighbor and bilinear interpolation techniques. While low-resolution simulation estimated using bilinear interpolation has the least error (RMSE and NRMSE of 1.753 °C and 0.016) but the highest bias (NMBF of 0.013), increasing the resolution reduced the bias greatly (NMBF lowered to 0.007) with no significant increase in error (Table 4)..


### 4.1.2 Relative humidity

The surface relative humidity in Brunei simulated using high-resolution and low-resolution simulations had a very good correlation coefficient (above 0.8). Although WRF_Chem_20km suffered slightly in correlation (by 0.017) compared to WRF-Chem_100km using nearest neighbor estimation technique, the overall performance was better using 20 km horizontal



resolution ( lower error and bias) with RMSE, NRMSE and NMBF of 7.544 %, 0.142 and -0.027 respectively. Also, the bilinear interpolation method maintained almost the same correlation for both resolutions and the 20 km resolution simulation had lower error and bias with RMSE, NRMSE and NMBF of 7.293 %, 0.138 and -0.024 respectively (see Table 2). This confirms that higher horizontal grid resolution improves the representation of surface relative humidity in our WRF-Chem simulations.

In the case of air profile relative humidity in Brunei, increased simulation resolution worsened the correlation and gave higher error especially with bilinear interpolation method (RMSE and NRMSE of 16.101 % and 0.186). Although high-resolution simulation with nearest neighbor estimation reduced the bias greatly (NMBF of 0.100), the same simulation with bilinear interpolation method gave the highest bias (NMBF of 0.125) (see Table 3).

Similarly, the air profile relative humidity in Singapore had its performance worsened with higher resolution simulation. The
correlation coefficient reduce from 0.844 in the low resolution simulation to 0.823 and 0.829 with nearest neighor and bilinear interpolation method respectively, in high-resolution simulation. Although bias was lowest with bilinear interpolation on high-resolution simulation, the errors were higher especially with the nearest neighbor estimation (RMSE and NRMSE of 15.281 and 0.193) (Table 4).

### 4.1.3 Wind speed

The surface wind speed in Brunei station underperformed with increased parametrization using high-resolution simulation. Using nearest neighbor calculation, WRF-Chem_20km simulation had lower correlation coefficient relative to WRF-Chem_100km simulation (0.329 compared to 0.469). Also, the error and bias for WRF-Chem_100km simulation were lower than WRF-Chem_20km simulation with RMSE, NRMSE and NMBF of 1.165 m/s, 0.226 and 0.019 compared to 1.221 m/s, 0.237 and 0.101, respectively. Although, bilinear interpolation technique improved high-resolution simulation results but the
performance (correlation, errors and bias) were lesser than the calculated results using nearest neighbor method with coarse simulation (see Table 2).

Considering air profile wind speed in Brunei, all the simualtions had a good correlation coefficient (above 0.8). Increased resolution simulation results had worse correlation coefficients but the bilinear interpolation estimation improved the correlation slightly. Overall, the high-resolution simulation gave lower error and bias compared to low-resolution simulation
(see Table 3).

In Singapore on the other hand, the air profile wind speed simulated using high resolution improved overall results. The correlation coefficient increased to 0.915 from 0.913 and 0.897 for bilinear interpolation and nearest neighbor techniques respectively. Also, the RMSE, NRMSE and NMBF reduced to 3.384 m/s, 0.114 and -0.093 respectively (see Table 4).



### 4.1.4 Wind direction

The transport of aerosol is mainly governed by wind and its direction. The wind direction at 10 meters above ground level using finer resolution had better representation than the low-resolution simulation. Using nearest neighbor method, The correlation coefficient for WRF-Chem_20km was slightly higher than WRF-Chem_100km (0.258 against 0.252). Also, the error and bias observed were lower in high-resolution simulation. WRF-Chem_20km had RMSE, NRMSE and NMBF of 85.082 degree, 0.243 and -0.051 compared to WRF-Chem_100km with 103.342 degree, 0.295 and -0.115, respectively. Overall, bilinear interpolation calculation method yielded better results for high-resolution simulation with RMSE, NRMSE and NMBF of 78.669 degree, 0.225 and -0.039 respectively (see Table 2).

In Brunei, air profile wind direction simulated using high-resolution simulation had worsened correlation but lowered bias compared to low-resolution simulation with either nearest neighbor or bilinear interpolation estimation. The bilinear interpolation calculation greatly improved the results especially for the low-resolution simulation with correlation coefficient of 0.775, and RMSE and NRMSE of 51.222 degree and 0.180 respectively (see Table 3).

Whereas in Singapore, wind direction profile simulated using high-resolution simulation yielded better correlation coefficent (0.681) with either bilinear interpolation or nearest neighbor methods. The bias was also reduced to -0.046 and -0.055 while estimating using nearest neighbor and bilinear interpolation techniques respectively. While the error was lowered with increased resolution simulation, the least error (RMSE and NRMSE of 66.541 degree and 0.208) was obtained when using bilinear interpolation calculation on coarse resolution simulation (see Table 4).

Overall, the model in this study simulated air profile up to 20 km altitude (in the upper troposphere and lower stratosphere region). The air temperature and wind speed profile in Brunei and Singapore had an overall improvement (lower errors and bias/higher correlation) with finer resolution simulation. Wind direction profile using high resolution simulation suffered a slight increase in errors, but with higher correlation in Singapore and lower bias in both Brunei and Singapore.

The high-resolution WRF-Chem simulations performed better in meteorology representation, though the low-resolution simulations results were also very good. Thus, the model developed and presented in this work proved to be an effective tool in simulating atmospheric conditions and deep convection in the South-East Asia region.

### 4.2 Air Quality Evaluation

The simulated aerosols were evaluated by comparing the results to the observed mean daily particulate matter concentrations in Singapore and Brunei Darussalam. Figures 4 and 5 show the comparison of the spatiotemporal distribution of particulate matter over South East Asia during the haze period using low and high-resolution simualtions. Also, Figures 6 and 7 show the





15-day simulated PM2.5 and PM10 values to the observations in Singapore and Brunei Darussalam between 15[th] and 30[th] June 2013, respectively.

The WRF_Chem_100km simulation gave a good result for PM2.5 levels in Singapore with slight underestimation over the 15 days but it largely underestimated the PM10 levels in Brunei. The correlation coefficients were 0.8411 and 0.1913 for PM2.5 in Singapore and PM10 in Brunei, respectively (Table 5) (Figures 6 and 7). The parameterization involved in this coarse simulation is relatively minimal and over-approximation of topography could have induced higher inconsistencies in the spatial distribution of particulate matter across the region.

In the case of the WRF-Chem_20km simulation, the simulated PM2.5 in Singapore had a correlation coefficient of 0.8864 and the simulated PM10 levels in Brunei had 0.3469 correlation coefficient (Table 5) (Figures 6 and 7). Despite that particulate matter levels were generally underestimated across the region in 20 km grid scale than 100 km grid scale simulation (larger bias, RMSE and NRMSE), the improvement in correlation showed that the higher horizontal resolution with 20 km grid scale had a better representation of PM.

Based on the current knowledge of processes (physical and chemical) and observations in estimating biomass burning emissions, many uncertainties still existed, for which enhancements were required and adopted in several models to scale the bottom-up inventories to the top-down constraints (Kaiser et al., 2012). Wu et al (2011) multiplied 3BEM OC and BC surface aerosol emissions by a factor of 5 for the 2006 fire season and Kaiser et al. (2012) enhanced the GFASv1.0 particulate emissions by a factor of 3.4 (Kaiser et al., 2012; Wu et al., 2011). In order to study this effect, the fire emissions were elevated by changing the default biomass burning emission enhancement factor from 1.3 to 6 in simulatng the June 2013 haze episode in the region, and the new results were compared with default coarse and finer resolution simulations (see Figures 4 and 5).

After enhancement of biomass burning emissions, the simulation (WRF-Chem_20kmX) gave a very good representation of particulate matter distribution across the South East Asia region. The PM10 levels in Brunei were adequately represented in this simulation with a correlation coefficient of 0.5974. Although PM2.5 levels in Singapore were slightly overestimated, the correlation coefficient was maintained at 0.8847 (Table 5) (Figures 6 and 7).

Moreover, it is evident from the comparison of the time-averaged total column aerosol optical depth at 550 nm of the simulations against reanalysis from MERRA-2 (M2T1NXAER v5.12.4) that the default factor of 1.3 for biomass burning emission in the model underestimated the aerosols and could not repeat the AOD reanalysis in both lower and higher horizontal grid scale simulations. On the other hand, the enhacement factor of 6 used in high resolution simulation fairly repeated AOD reanalysis with positive bias and overestimation of the aerosols especially across the highly polluted zones in the South East Asia region (see Figure 8). From this analysis,it could be suggested that an optimum enhancement factor for fire emissions to simulating aerosols in this region should be above 1.3 and below 6.



The biomass burning emissions from Sumatra was very intense in June 2013 and it was mainly responsible for the haze and
increase in PM concentrations in Singapore. It was observed that the fire emissions elevated around the eastern part of Sumatra
on the 19th of June 2013 which is in correlation with the increased frequency of hotspots captured by MODIS.  The winds
blowing in the northeast and easterly direction over Sumatra and Peninsular Malaysia through the South China Sea is mainly
responsible for the long-range transport of particulate matter to Singapore and Borneo region from the biomass burning sources
in Sumatra. Some sparsely spread fires were also detected in the Sarawak region in Borneo and they had more influence on air
pollution levels in Brunei during this period. This spatial distribution was clearly seen and evident from the simulations with
20 km horizontal resolution than 100 km (Figures 4 and 5).The elevated fire emissions and air pollution level in this region
persisted until the 23rd of June when it peaked and subsequently started its decline. The wind direction gradually shifted around
23rd June from northeasterly direction to the north over the South China Sea, while westerly winds started blowing from
eastern Borneo and weak winds prevailed over the Borneo region. The decline in fire emissions and change in wind direction
could be responsible for the decline in particulate matter levels observed in Brunei Darussalam after this period.

**5 Summary and Conclusion**

We have simulated the particulate matter distribution from biomass burning emissions across South East Asia during the June
2013 haze event. We synergistically leveraged on hot-spot datasets across the region from MODIS and VIIRS and assessed
the effect of horizontal grid scales and enhancement factor for biomass burning emissions for particulate matter simulations.
The result of the statistical metrics for the model simulations against the observation in Brunei and Singapore revealed that the
model simulated the meteorology (surface and vertical air profile for variables such as temperature, relative humidity, wind
direction and wind speed) adequately across the region.

It was also confirmed from the analysis that higher resolution simulations gave a better representation of some prognostic
meteorological variable while some underperformed. Interpolation method for obtaining simulation results was also varied
between nearest neighbor and bilinear interpolation. Considering surface meterology evaluations using nearest neighbor
method, higher resolution yielded lower bias and errors in all meteorological variables assessed except wind speed. On the
other hand, increased resolution simulation resulted in lesser correlation in all variables except wind direction.

On the other hand, bilinear interpolation performed better than nearest neighbor technique for surface meteorology at Brunei
in higher and lower model resolutions simulations except wind speed. It proved to be an efficient method for estimating station
variables such as surface temperature on low resolution simulation and it gave overall best performance in obtaining surface
relative humidity and wind direction in Brunei station. However, nearest neighbor extraction seemed to yield better result for
wind speed in Brunei.



The fire emissions in Sumatra was very intense in June 2013 and it was mainly responsible for the haze and increase in PM concentrations in Singapore during this period. While these emissions contributed to the particulate matter transported across the region northeasterly by winds, the less intense biomass burning across Borneo, such as in Sarawak, had more impact on the levels of transboundary particulate matter received in Brunei region.

Furthermore, comparison of the observed and model simulated PM2.5 and PM10 emissions between the 15th and 30th June 2013 in Singapore and Brunei Darussalam respectively, showed that the model was able to capture the particulate matter emissions during the haze episode in South-East Asia, though with huge underestimation in regions such as Borneo (Figures 6 and 7). The underestimation in particulate matter levels could be taken care of by adopting biomass burning emissions enhancement in the model. This requirement is evident from the comparison of the time-averaged total column aerosol optical depth at 550 nm among the simulations as compared with MERRA-2 reanalysis in the South East Asia region (see Figures 7). Therefore, we can conclude that the model succeeded in capturing the particulate matter emissions across the region during the haze episode but had a large underrepresentation in some parts such as the Borneo region. The discrepancies observed may be as a result of errors present in the biomass burning emission inventories. Despite leveraging on VIIRS to augment MODIS hot-spot data, it was very likely that some burning hot-spots were not captured due to cloud cover and vegetation canopies around the Borneo region.

In future research, improvement in the simulation could be made by employing finer emission inventory and ground measurement of biomass burning emissions for WRF-Chem model emissions input to overcome the undetected emissions from satellites. Also, to improve the particulate matter representation, the emissions in the model require enhancement. A biomass burning enhancement factor above 1.3 and below 6 could be included to bring the bottom up inventories to the observed values. Furthermore, discrepancy caused by the underrepresentation of the model resolutions should be avoided by adopting a finer horizontal resolution in resolving the small-scale circulation and the cloud convection while considering prognostic variables staggering on the grid. These could improve significantly the understanding of particulate matter distribution from biomass burning emissions across the region.

**Author Contributions**

ARA, with support from LD, conducted the simulations, processed the data, and prepared the data visualization with contributions from all authors. LD, MIP, and LCD sourced for all observation datasets required. ARA prepared the paper and it was reviewed by LD, MIP, LCD and ZT.



**Competing Interest**

The authors declare that they have no conflict of interest.

**Code Availability**

WRF-Chem    is    an    open-source    community    model.    The    source    code    is    available    at
http://www2.mmm.ucar.edu/wrf/users/download/get_source.html (last access: December 2018).

**Acknowledgments**

We would like to thank the WRF-Chem developers for their support in setting up the model. The authors would like to thank
the Brunei Darussalam Meteorological Department (BDMD) and the Department of Environment, Parks and Recreation
(JASTRe) for providing meteorological and emission datasets used in this work. The corresponding author gratefully
acknowledges the support and funding (Universiti Brunei Graduate Scholarship) provided by Universiti Brunei Darussalam.
Also, we acknowledge NASA's Goddard Earth Sciences Data and Information Services Center (GES DISC) for the
dissemination of MERRA. Visualizations and analysis in this paper were made using the NCAR Command Language while
satellite observations were produced with Giovanni online datasystem, developed and maintained by the NASA GES DISC.

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





**Table 1: Observation Locations and data availability at each location**

| Location | Latitude | Longitude | Species Measured/Observed |
|---|---|---|---|
| Brunei Darussalam | 4.93 | 114.93 | Daily PM10, surface and vertical air temperature, relative humidity, wind speed and direction. |
| Singapore | 1.22 | 103.59 | Daily PM2.5, vertical air temperature, relative humidity, wind speed and direction. |

**Table 2: Correlation coefficients, NRMSE, RMSE and NMBF for temperature at 2 m, surface rel. humidity at 2 m, wind speed and direction at 10 m for WRF-Chem simulations (100km and 20km) at Brunei station (4.93º N, 114.93º E) between 15th and 29th June 2013.**

| | | WRF-Chem_100km | | WRF-Chem_20km | |
|---|---|---|---|---|---|
| | | Nearest Point | Bilinear Interp. | Nearest Point | Bilinear Interp. |
| Temperature | Correlation coef.(r) | 0.924 | 0.929 | 0.908 | 0.912 |
| (Celsius) | NRMSE | 0.191 | 0.127 | 0.154 | 0.137 |
| | RMSE | 2.175 | 1.444 | 1.76 | 1.561 |
| | NMBF | -0.066 | -0.028 | -0.039 | -0.034 |
| Relative | Correlation coef.(r) | 0.86 | 0.853 | 0.843 | 0.852 |
| Humidity | NRMSE | 0.212 | 0.182 | 0.142 | 0.138 |
| (%) | RMSE | 11.254 | 9.665 | 7.544 | 7.297 |
| | NMBF | 0.096 | 0.076 | -0.027 | -0.024 |
| Wind Speed | Correlation coef.(r) | 0.469 | 0.421 | 0.329 | 0.374 |
| (m/s) | NRMSE | 0.226 | 0.253 | 0.237 | 0.240 |
| | RMSE | 1.165 | 1.304 | 1.221 | 1.233 |
| | NMBF | 0.019 | 0.246 | 0.101 | 0.180 |
| Wind Direction | Correlation coef.(r) | 0.252 | 0.278 | 0.258 | 0.304 |
| (degree) | NRMSE | 0.295 | 0.248 | 0.243 | 0.225 |
| | RMSE | 103.342 | 86.835 | 85.082 | 78.669 |
| | NMBF | -0.115 | -0.049 | -0.051 | -0.039 |




**Table 3: Correlation coefficients, NRMSE, RMSE and NMBF for air temperature, relative humidity, wind speed and direction of WRF-Chem simulations (100km and 20km) averaged for Brunei between 15th and 29th June 2013.**

| | | WRF-Chem_100km | | WRF-Chem_20km | |
|---|---|---|---|---|---|
| | | Nearest Point | Bilinear Interp. | Nearest Point | Bilinear Interp. |
| Temperature | Correlation coef.(r) | 0.999 | 0.999 | 0.999 | 0.999 |
| (Celsius) | NRMSE | 0.017 | 0.017 | 0.017 | 0.016 |
| | RMSE | 1.852 | 1.792 | 1.775 | 1.759 |
| | NMBF | -0.013 | -0.014 | -0.009 | -0.011 |
| Relative | Correlation coef.(r) | 0.858 | 0.869 | 0.849 | 0.817 |
| Humidity | NRMSE | 0.171 | 0.163 | 0.176 | 0.186 |
| (%) | RMSE | 14.969 | 14.271 | 15.283 | 16.101 |
| | NMBF | 0.124 | 0.109 | 0.100 | 0.125 |
| Wind Speed | Correlation coef.(r) | 0.826 | 0.830 | 0.817 | 0.819 |
| (m/s) | NRMSE | 0.145 | 0.148 | 0.142 | 0.142 |
| | RMSE | 4.263 | 4.405 | 4.193 | 4.217 |
| | NMBF | -0.136 | -0.119 | -0.088 | -0.099 |
| Wind Direction | Correlation coef.(r) | 0.757 | 0.775 | 0.737 | 0.744 |
| (degree) | NRMSE | 0.203 | 0.180 | 0.191 | 0.184 |
| | RMSE | 57.753 | 51.222 | 53.715 | 52.060 |
| | NMBF | 0.066 | 0.056 | 0.053 | 0.049 |







**Table 4: Correlation coefficients, NRMSE, RMSE and NMBF for air temperature, relative humidity, wind speed and direction of**
**WRF-Chem simulations (100km and 20km) averaged for Singapore between 15th and 29th June 2013.**

|  |  | WRF-Chem_100km | | WRF-Chem_20km | |
|---|---|---|---|---|---|
|  |  | Nearest Point | Bilinear Interp. | Nearest Point | Bilinear Interp. |
| Temperature | Correlation coef.(r) | 0.999 | 0.999 | 0.999 | 0.999 |
| (Celsius) | NRMSE | 0.016 | 0.016 | 0.017 | 0.016 |
|  | RMSE | 1.780 | 1.753 | 1.834 | 1.772 |
|  | NMBF | -0.010 | -0.013 | -0.007 | -0.007 |
| Relative | Correlation coef.(r) | 0.844 | 0.844 | 0.823 | 0.829 |
| Humidity | NRMSE | 0.186 | 0.182 | 0.193 | 0.187 |
| (%) | RMSE | 14.705 | 14.418 | 15.281 | 14.793 |
|  | NMBF | -0.178 | -0.153 | -0.141 | -0.100 |
| Wind Speed | Correlation coef.(r) | 0.897 | 0.913 | 0.915 | 0.915 |
| (m/s) | NRMSE | 0.123 | 0.117 | 0.114 | 0.115 |
|  | RMSE | 3.613 | 3.444 | 3.384 | 3.435 |
|  | NMBF | -0.113 | -0.123 | -0.093 | -0.105 |
| Wind Direction | Correlation coef.(r) | 0.597 | 0.630 | 0.681 | 0.681 |
| (degree) | NRMSE | 0.224 | 0.208 | 0.215 | 0.215 |
|  | RMSE | 71.523 | 66.541 | 68.712 | 68.712 |
|  | NMBF | -0.079 | -0.086 | -0.046 | -0.055 |

**Table 5: Correlation coefficients, NRMSE, RMSE and NMBF for WRF-Chem simulations (100km and 20km) of PM2.5 (Singapore)**
**and PM10 (Brunei) between 15th and 29th June 2013.**

|  | WRF-Chem_100km | | | | WRF-Chem_20km | | | | WRF-Chem_20kmX | | | |
|---|---|---|---|---|---|---|---|---|---|---|---|---|
|  | Correlation coef (r) | RMSE ($\mu g/m^3$) | NRMSE | NMBF | Correlation coef (r) | RMSE ($\mu g/m^3$) | NRMSE | NMBF | Correlation coef (r) | RMSE ($\mu g/m^3$) | NRMSE | NMBF |
| Singapore PM2.5 | 0.841 | 92.852 | 0.412 | -1.801 | 0.886 | 116.735 | 0.518 | -5.827 | 0.885 | 113.023 | 0.501 | 0.694 |
| Brunei PM10 | 0.191 | 65.641 | 0.721 | -4.761 | 0.347 | 68.196 | 0.749 | -6.514 | 0.597 | 26.326 | 0.324 | 0.180 |


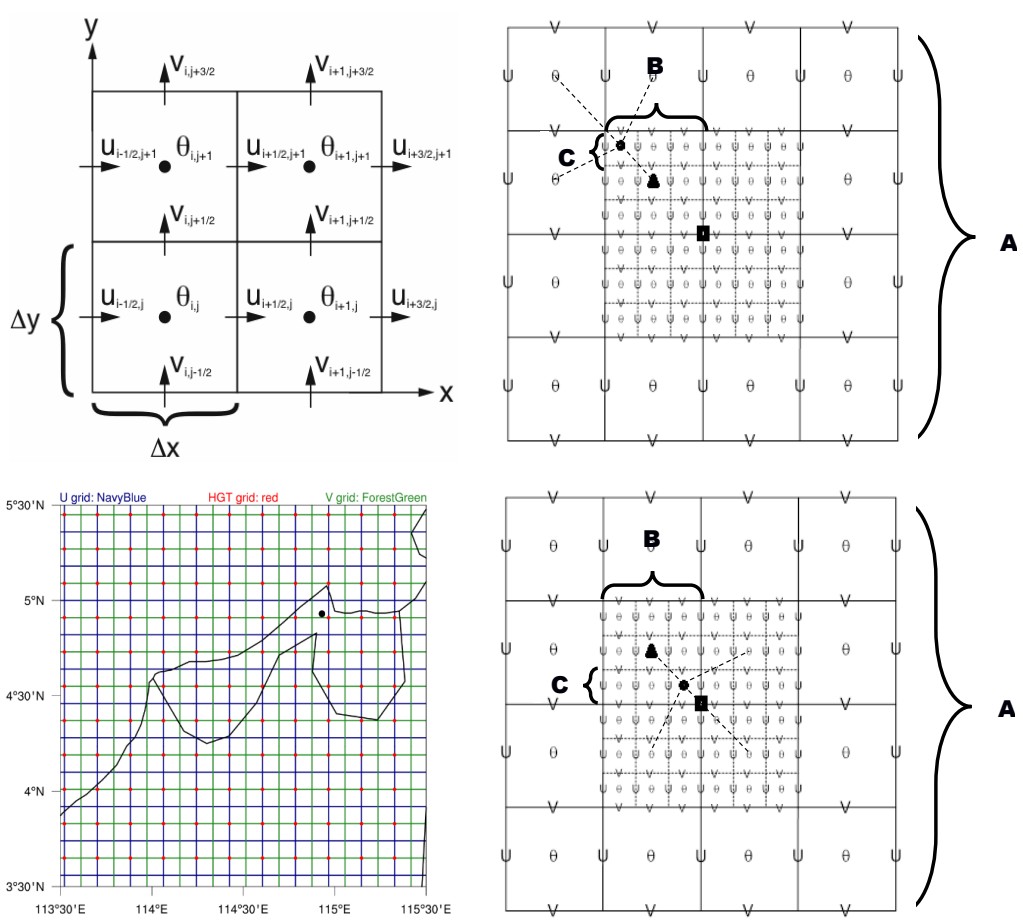

**Figure 1: Horizontal model resolution versus interpolation in WRF-Chem grid staggering. Horizontal Arakawa staggering in WRF-Chem** (Arakawa and Lamb, 1977) **– top left; Nearest point and bilinear interpolation (case A) – top right; Brunei station (4.93º N, 114.93º E) relative to variables staggering – bottom left and; Nearest point and bilinear interpolation (case B) – bottom right.**

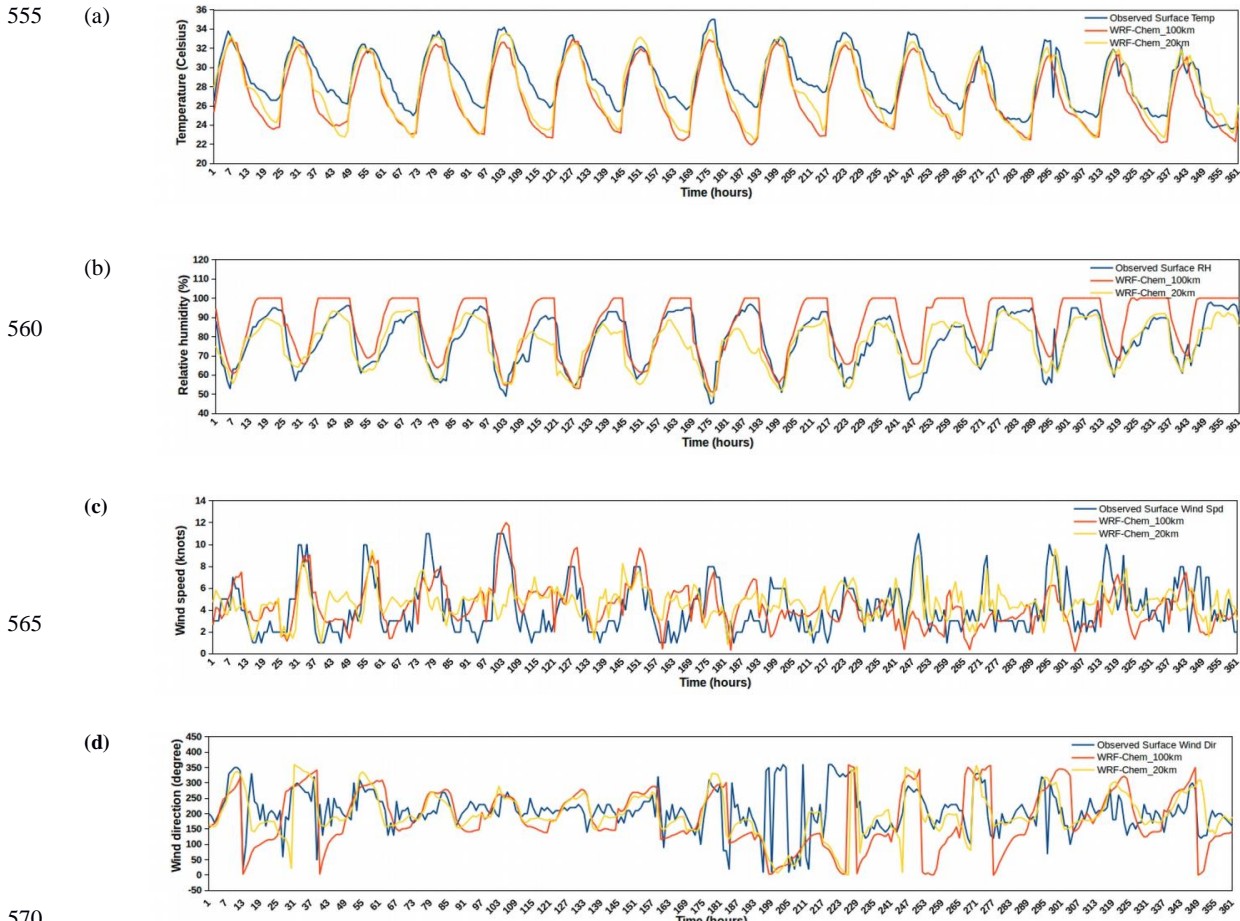

(a)




**Figure 2:** Time-series of observations and WRF-Chem simulations for (a) surface temperature (T at 2 m in Celsius); (b) surface relative humidity (RH at 2 m in %) (c) 10 m wind speed (knots) and; (d) 10 m wind direction (degree), at Brunei International Airport

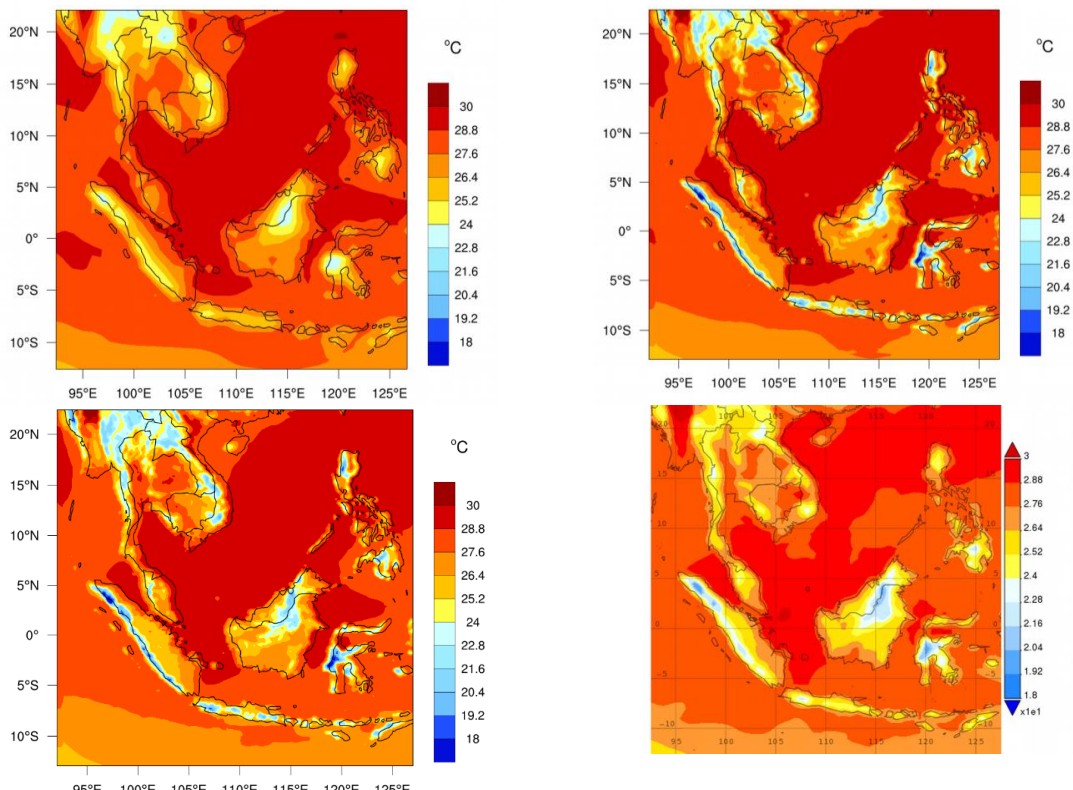

**Figure 3: Time Averaged surface temperature (2 m) in South-East Asia between 15ᵗʰ and 29ᵗʰ June 2013. WRF-Chem_100km resolution – top left; WRF-Chem_20 km resolution – top right; WRF-Chem_20 kmX resolution – bottom left and; MERRA-2 reanalysis – bottom right.**






**Figure 4: Particulate matter (PM10) WRF-Chem simulations at 00UTC on 16th, 20th, 24th and 28th of June 2013 in South-East Asia (WRF-Chem_100km - left column, WRF-Chem_20km – middle column and WRF-Chem_20kmX – right column)**



**Figure 5: Particulate matter (PM2.5) WRF-Chem simulations at 00UTC on 16th, 20th, 24th and 28th of June 2013 in South-East Asia (WRF-Chem_100km - left column, WRF-Chem_20km – middle column and WRF-Chem_20kmX – right column)**





(a)


(b)


(c)


**Figure 6: WRF-Chem simulated and observed PM2.5 in Singapore (a) WRF-Chem_100km; (b) WRF-Chem_20km and; (c) WRF-Chem_20kmX**


(a)


(b)


(c)



**Figure 7: WRF-Chem simulated and observed PM10 in Brunei (a) WRF-Chem_100km; (b) WRF-Chem_20km and;**
**(c) WRF-Chem_20kmX**

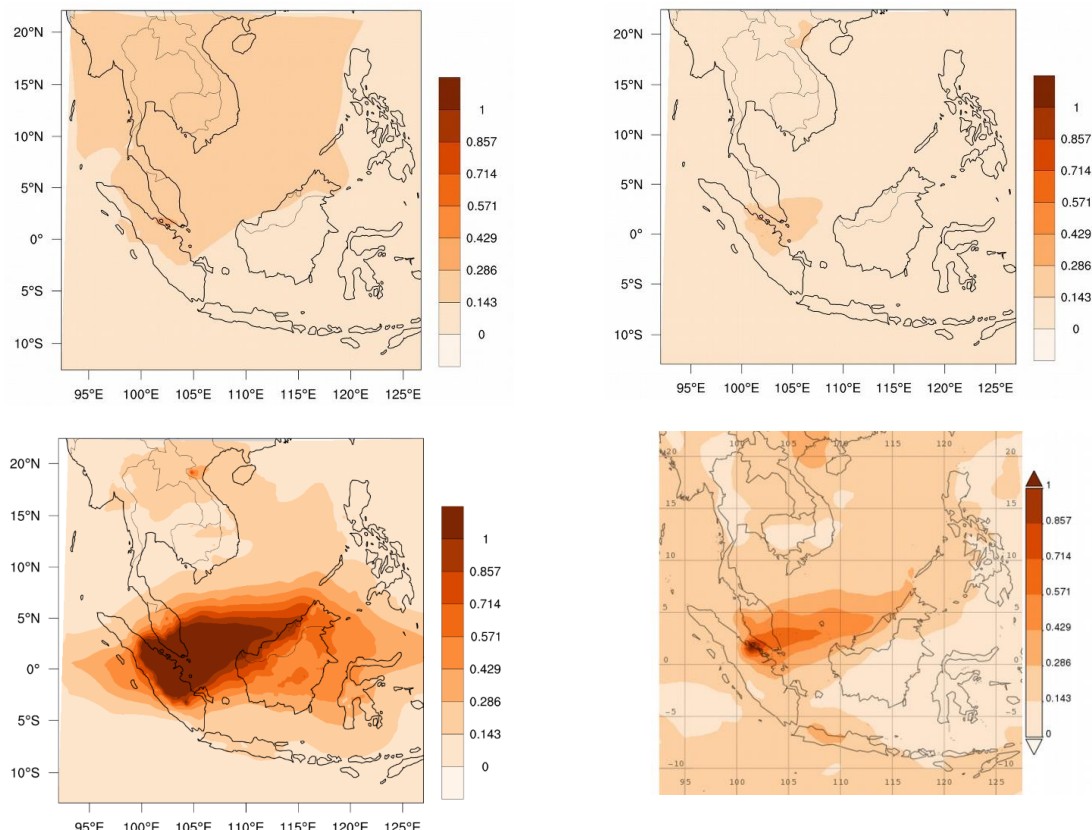


**Figure 8: Time-Averaged Total Aerosol Optical Depth in the column at 550nm simulated over South-East A between 15th and 29th June 2013. 100km resolution – top left; 20km resolution – top right; 20kmX resolution – bottom left and; MERRA-2 reanalysis – bottom right**