# Peer review of "Sensitivity of WRF-Chem model resolution in simulating particulate matter in South-East Asia"

_Atmospheric Chemistry and Physics, 2019_

## Referee Comment (RC1) · Anonymous Referee #5 · 7 Apr 2020

This paper uses the WRF and WRF-Chem model to simulate the severe haze events in June 2013 in the SE Asian region. It calculates the biomass burning emissions using the 3BEM model with inputs from the MODIS and VIIRS fire "hot spots" and the WFABBA database while takes other emission inventories for anthropogenic and biomass burning emissions. Model simulations with two horizontal spatial resolutions, 100-km and 20-km, are conducted; results of a few meteorological fields and concentrations of PM2.5 and PM10 are compared with surface measurements at two ground stations and model calculated AOD is compared with the reanalysis product from MERRA-2. It concludes (a) the model is adequate, (b) the 20-km resolution results are somewhat better than the 100-km resolution counterparts, and (c) the biomass

burning emissions should be enhanced by a factor of 1.3 to 6.

This paper presents a routine model evaluation instead of a thorough analysis, which is fine and a needed exercise. However, I have several concerns of the model evaluations and think a major revision is necessary.

1. Purpose: From the introduction, it seems that the present study is motivated by the recent MICS-Asia Phase III study pointing out that most air quality model were developed mostly in Europe and USA such that the parameterizations and assumptions may not be suited for Asia, and calling for "rigorous investigations on the performance, sensitivities and uncertainties of these models in Asia". However, the present study does not test any of the parameterizations and assumptions in the WRF or WRF-Chem model, but only choose to examine the spatial resolutions.

2. Sensitivity: the title suggests that the major focus is to address the sensitivity of the model performance to the model spatial resolution. But at the end, the model results show little sensitivity to the spatial resolutions. Rather, the emission amount dictates the simulation quality. This should be explicitly quantified.

3. Model resolution: 100-km and 20-km are very coarse resolutions for a regional model. Most global models are using similar resolutions nowadays. 20- and 4-km (or finer) resolutions would make a much better sense.

4. Biomass burning emission: Because the biomass burning emission is such a key parameter determining the model simulation quality, it grants a much more thorough assessment in the study. The description of emission needs to be in more detail. For example, what are the input quantities used in 3BEMS? How the MODIS, VIIRS, and WFABBA datasets are implemented in 3BEMS? What species are emitted? And finally, how does the biomass burning emission calculated in the present work compared to other readily available emission datasets, such as GFED4, GFAS, FEER, FIN, etc. for the same regions during the same time period?

5. Model evaluation: The evaluations are carried out mainly by comparisons of data from two ground stations, one in Singapore and one in Brunei. The model results from two different spatial resolutions and two different interpreting methods (nearest neighbor and bilinear) are evaluated using correlation coefficients, NRMSE, RMSE, and NMBF and resented in Table 2, 3, 4, and 5. Given the slight differences of these parameters between the results from 100-km and 20-km resolutions or between the two interpolation methods, the question is: are these difference significant enough that you can conclude that one resolution or one interpolating method is definitely and consistently better than the other one? Also, what lacks is an overall "skill score" that considers all the above quantities, which is commonly used in the weather and air quality forecast communities.

Specific comments:

Page 2, line 43: "numerically studied" – what does it mean?

Page 2, line 44-45: "The correlation coefficient" – between what?

Page 4, line 118: "...could be used" – did you actually use the daily fires detected from different satellites? If so, it should be "...were used".

Page 4, line 126-128: How did you use the two sets of emissions (RETRO and EDGAR)? Did you average them?

Page 4-5, line 133-134: What is "NALROM" simulation? Why do you have to use the "idealized" northern hemispheric mid-latitude and clean environment conditions, which has nothing to do with your study area?

Page 5, line 139: "The emission inventories were updated for each day": How do you update the emission inventories for each day for anthropogenic? I don't think the RETRO or EDGAR provide daily emissions.

Page 6, line 169-174: I find these paragraphs and associated Figure 1 are confusing and seem unnecessary. Why assuming a series of grid cells with different horizontal

resolution is necessary while your model uses even grid space? To me, just describe briefly how the evaluation of model output in a grid with a point measurement was done is enough.

Page 7-9, subsection 4.1.1-4.1.4 and Table 2-4: The organization of section 4.1.1-4.1.4 and Table 2-4 should be consistent. At the present, the Tables are organized as one table per station with all four met variables, whereas the subsections are organized as one met variable per subsection at all stations. This makes the readers going back-and-forth with the tables while reading the text. I suggest reorganize either the subsections or the tables to make flow together.

Page 7, line 209 and Figure 2: Figure 2 shows all four met variables, but it is only mentioned temperature here, not in the other subsection.

Page 7, line 214 and Figure 3: Figure 3 also shows 20kmX and MERRA-2, but they are not mentioned here! Why are they used in Figure 3?

Page 9, line 282-283: The statistics of evaluation from the 100- and 20-km simulations are very comparable according to the numbers listed in Table 2-4. Repeating what I said at the beginning, there should be overall skill scores to quantify the significance of the differences.

Page 9, line 283: "…also very good": What is the standard of being "very good"? These subjective words should be avoided in the model evaluation. Instead, quantitative information should be given, such as "within a factor of xx%".

Page 9, line 287, Figures 4 and 5: I suggest swap Figure 4 and 5, i.e., show PM2.5 first in Figure 4 and then PM10 in Figure 5 to be in the same order of Figure 6 and 7.

Page 10, line 291: Again, please avoid using subjective phrases such as "good results".

Page 10, line 294: What is "over-approximation of topography"? How was it done? It is very unclear.

Page 10, line 299-300: But the 20-km simulation has larger bias than the 100-km simulation. Why is it better? R is not everything. In many cases, bias is as or more important.

Page 10, line 304, GFAS: Why is GFAS relevant here since you don't use GFAS?

Page 10, line 308: another "very good". Please be quantitative and objective.

Page 10, line 309-310: What is the size range of biomass burning aerosols? Any fraction in the coarse mode with diameters greater than 2.5 um?

Page 10, line 311, regarding Table 5 and Figure 7: Table 5 shows that 20kmX simulation resulted in a positive bias (0.180) for PM10 in Brunei, but Figure 7 (bottom panel) seems to suggest an overall negative bias. Please check.

Page 10, line 313: Describe MERRA-2. It just suddenly shows up here without any description. In fact, as I said earlier, Figure 3 showed a figure from MERRA-2 without any context.

Page 10, line 317-318: a factor of 1.3-6 is a very large range. Any constraint? It seems your study suggest a factor 6 since the default 1.3 does not work for PM2.5 and PM10; on the other hand, using a factor of 6 will substantially overestimate the AOD compared to MERRA-2.

Page 11, line 340: What is the criterium for "adequately"? Be quantitative, such as "within a factor of x".

Page 12, line 361-362: "...without fire emission enhancements, the model succeeded in capturing PM emission across the region": How did you reach that conclusion? You only evaluated the PMs at two stations, not many stations across the region.

Page 12, line 363: "...may be as a result of errors present in the biomass burning emission inventories": It is important to compare the biomass burning emission from this work to other emission datasets to at least see how they compare.

Page 12, line 369: It is premature to suggest an enhancement factor of 1.3-6 used in this particular work for all emission inventories, unless you have compared all the available ones.

---

## Referee Comment (RC2) · Anonymous Referee #1 · 17 Apr 2020

General comments:

The manuscript has applied the WRF-Chem model to study the biomass burning haze episode in Southeast Asia from 15th to 30th June 2013, with research focus on grid resolution and interpolation approach. The research scope of the manuscript fits well with the interest of the readership of ACP journal. However, several improvements need to be made before the manuscript is deemed suitable for publication.

The sensitivity analysis of the model performance is the main highlight of the work, with focus on the grid resolution test. However, 20 km used as the fine grid resolution is comparatively coarse compared to previous work done in the region that have done

into 9 km grid size. Given sufficient computational ability, 20 km can be better applied as the nest to the coarse grid (100 km) which will be a better option for more accurate modelling performance. For the sensitivity study to be conducted in this rather short period of study, the number of verification stations and attempted physical settings are insufficient. On top of that, the scientific basis on the grid resolution selection and enhancement ratio is lacking to convince on the applicability of such method. The limitation of these method should also be highlighted in the manuscript. In order to improve the performance and reliability of the output data, it is advisable to test and choose the grid settings wisely to achieve desired goal.

The paper has highlighted several aims that are not attained especially aim 1 and 2 that are briefly discussed with subjective statement from the modelling output. Missing references and syntax error are easily sighted in the manuscript. Detailed and specific comments are given as below. Therefore, the manuscript needs to go through major revision before it is ready for acceptance and publication.

Specific comments:

1. L57-59: Perhaps you can discuss from the point of view of emission composition from the burning materials

2. L83-39: Case June 2013 were studied in Oozeer et al. (2016) (cited in manuscript) with a higher resolution of 9 km and the model performance is reasonably well. How do you argue that the 20 km setting in current work is known as high resolution compared to theirs? Also, with 100 km to 20 km, it is already possible that you are able to have a mother domain (100 km) with nest (20 km), this is able to further improve your weather prediction result seeing that you are using a $1°x1°$ ($\sim$100 km) NCEP FNL dataset (L111-114).

3. L92-107: Detailed description of the options are not necessary, instead please state the reason and basis why these chemistry and physics settings are chosen.

4. Please include Section 3.1 into Section 2 because they are part of the WRF-Chem settings.

5. L125-127: Can you please cite this statement and explain why you are still using the WFABBA dataset if it is inaccurate? If not, please kindly explain what improvement have been made to tackle this issue.

6. L188-200: This section should be moved to Section 3.2.

7. L218-221: The model performance is less satisfactory, especially for temperature that could be easily meet the standard of RMSE belowïĂăïĆś1.5ïĆřC in the stable weather condition in tropics. The performance of predicted weather variables have a big impact on the chemistry field, so please kindly consider using the nesting option and nudging function to improve the result for the weather field in the first place.

8. L250-289: Same applies for humidity and wind speed. The verification result of upper layer using radiosonde is missing from the content.

9. L310-312: Please explain how to obtain 6 as a factor and what do you based on?

10. L306-312: Previous work conducted in the region has suggested that the fire injection height in the region has to be adjusted. Please kindly refer to Wang et al. (2013) Atmospheric Research 122, 486–503.

11. L320-323: The argument is less convincing for just testing with one of the values, please kindly test with more settings to confirm that the enhancement factor are selected appropriately.

12. L325-335: The "formation" and "deep convection" aspects of the biomass burning emissions are totally not discussed in the result and analysis.

13. Section 4.2: Figure 4,5,6,7,8 showed that the enhancement factor for 20km improved the prediction because they produced larger amount of PM2.5. If comparing 100 km and 20 km, 100 km has also produced larger emission of PM2.5, hence, why

100 km case doesn't give a much better result compared to 20 km case?

14. Figure 1,3,4,5,8: Please properly label a,b,c,etc. next to each figures and describe them in the figure caption.

Technical corrections: (only listed a few)

1. L24: correct progronostic to prognostic

2. L29: correct enhacement to enhancement

3. L49: Make sure the citation is done properly for both microphysics and boundary layer schemes.

4. L92: Remove "list references"

5. L139: Missing reference for Mozart boundary condition.

6. L216: correct time-avergaed to time-averaged, distrubtion to distribution

---

## Referee Comment (RC3) · Anonymous Referee #2 · 19 Apr 2020

General Comment This study assessed the ability of WRF-Chem in capturing the spatial variability and concentrations of particulate emissions during haze 2013. PM10 data from Brunei and PM2.5 data from Singapore were used for model validation. PM10 concentration in Brunei had a correlation coefficient around 0.4, and the simulated PM2.5 level in Singapore had correlation coefficient around 0.9. Overall the study is interesting and contribute to the new knowledge the prediction of PM using WRF-Chem Model. To improve the manuscript, the need to include information on why they choose haze in 2013 for their study. Southeast Asia experienced several longer haze episodes other than haze in 2013. The also may need to explain on only two stations were used for PM validation with each of them has different parameter (PM10

and PM2.5). The authors need to improve their reference style of writing in the main text. I encourage them to include more recent papers on haze episode from Southeast Asia.

Detail Comment The study is for Southeast Asia. Why data only compared with PM10 in Brunei and and PM2.5 in Singapore in 2013? Line 46-47: Oozeer et al. (2016) numerically studied and analyzed convective mechanisms responsible for the uplift and transport of the particulate emissions from Sumatra over to Peninsular Malaysia during the 2013 event (Oozeer et al., 2016) - Oozeer et al. (2016) was mentioned two times in one sentence. Line 49: Name the "other station" mentioned in the sentence. Line 51: Gao and co-workers: Goa et al (2018)? Line 59: The reference given is quite old and not even mention PM2.5. There are many other studies related to PM10 and and PM2.5 in Southeat Asia after 1998, especially during haze episode. Among others are Amil et al. (Amil et al. 2016); Yin et al. (2019) Pimonsree et al. (2018) and many others. Amil, N., Latif, M. T., Khan, M. F. & Mohamad, M. 2016. Seasonal Variability of Pm2.5 Composition and Sources in the Klang Valley Urban-Industrial Environment. Atmospheric Chemistry and Physics 16(8): 5357-5381. Pimonsree, S. & Vongruang, P. 2018. Impact of Biomass Burning and Its Control on Particulate Matter over a City in Mainland Southeast Asia During a Smog Episode. Atmospheric Environment 195, 196-209. Yin, S., Wang, X., Zhang, X., Guo, M., Miura, M. & Xiao, Y. 2019. Influence of Biomass Burning on Local Air Pollution in Mainland Southeast Asia from 2001 to 2016. Environmental Pollution 254 Line 60: Include year after Tan et al. Line 65: Include year after Tie et al. Line 66: I am quite confused with the paragraphing starting from this line. "This study" refers to which study? I suggest the author to rewrite the main questions and aims of their study in a paragraph. Line 155-160: Is there any particular reason on why only two ground measurements (PM10 in Brunei and PM2.5 from Singapore) were used to evaluate the model performance of the simulations in this study? Why not the authors used other data available in Southeast Asia?

---

## Referee Comment (RC4) · Adedayo Rasak Adedeji et al. · 20 Apr 2020

This study deals with the "Sensitivity of WRF-Chem Model resolution in simulation particulate matter in South-East Asia". In other words, the topics focus on resolution and its impact on the PM simulation. However, this study just using two quite coarse resolutions 100 km and 20 km to discuss and evaluate the performance at two typical measurement sites. I will say the discussion and scientific results are quite poor and nothing new. Furthermore, the authors list four purpose of this paper aims in first section (L69-75), however, they do not really touch the points and detail discuss in this paper. Overall, the presentation skills are quite poor. It is like a report and a paper. I just list some detail comments in the following.

[Figure]

1. L69-70 "To simulate the formation, deep convection and long-range transport of the biomass burning emissions that resulted in higher particulate matter levels over the South-East Asia region." Do you think the resolution 20 km is good enough to discuss " deep convection" ? Where you discuss "the formation, deep convection and long-range transport of the biomass burning emissions. . .." in this paper ?

2. L71-72 "To identify the meteorology that caused and intensified the transport. . .." Where are the evidences you really discuss the linkages during the haze episode ?

3. L73 From Table 1 to Table 5, Actually, the performance between 100 km and 20 km are similar. How is the representative of this two stations in your study? Do you think these two different resolutions already approach your point list here " To analyze the response of meteorology and particulate matter simulation to horizontal grid resolutions "

4. L74-75, For me, it just a sensitivity test for the emission uncertainty in the simulation. How do you get the enhance factor "1.3" ?

5. The presentation skill for each figure is quite poor, you should give an order such as (a), (b), (c). . ., in each figure. Otherwise, it is difficult to follow up.

6. In Figure 3, you just put a Figure 20kmx, but does not discuss.

7. L287-289 "The high-resolution WRF-Chem simulations performed better in meteorology representation, though the low-resolution simulations results were also very good. " How do you think the results of the resolution 20 km and 100km "the simulation were also very good" ?

8. L 313-315 " After enhancement of biomass burning emissions, the simulation (WRF-Chem_20kmX) gave a very good representation of particulate matter distribution across the South East Asia region" For me, it seems just a tuning work for the emission in this case study.

---

## Referee Comment (RC5) · Anonymous Referee #3 · 21 Apr 2020

This study examined the sensitivity of the aerosol simulation in WRF-Chem to the horizontal resolutions (100km vs. 20km) in terms of meteorological fields and biomass burning emission amount in South-East Asia. The research itself is interesting, but the writing and structure need a major improvement, and so the understanding to the results.

Major comments:

1. You may need to specify "horizontal resolution" in the title. The model simulation like convection of emissions to a higher altitude and wider spatial scale largely depends on vertical resolution as well. Thus, the benefit of increasing horizontal resolution might

be limited by keeping vertical resolution the same as done in this study.

2. To my experience, a model often works well with a (or a few) certain spatial resolution because some parameters are tuned according to this spatial resolution when this model is developed. I am wondering if it is your case. Make sure that you are aware any these kinds of implicate settings, which can better our understanding of the complicated results.

3. Does the spatial resolution of your biomass burning (BB) emission change with the model spatial resolution, 20km vs. 100km?

4. Are the meteorological fields simulated freely or nudging towards reanalysis?

5. I think that you need to have ensemble runs for a robust result.

6. Based on your figures 6-8, increasing the biomass burning emission will help to bring the PM concentration level closer to the observation at Brunei only, but overestimate PM in Singapore and overestimate the overall AOD. Thus, increasing the BB emission seemingly doesn't help much. Or the factor of 6 is too large?

7. In the end, what lesson you learned with the limitation (e.g., evaluation with only two sites in the downwind region) and what suggestion would you give to others?

8. Do you have an explanation about the differences shown in these two sites?

9. The structure needs reorganization: In the section 2, you should add a sub-section to introduce your three experiments, especially the 25kmX, which starts to appear in the Figure 3 without any introduction. The grid staggering in your Section 4.1 belongs to an individual section for the method, not the section 4 which shows your results.

10. Can you label the multiple panels in a figure as (a), (b), (c). . .as you did in Figure 2? And it would be also very helpful to show a brief name in each panel, e.g., (a) WRF-Chem_100km.

Specific comments:

Line 22: change the 10 and 2.5 to subscript here and across the entire manuscript.

Line 24: remove the space between "quality" and "but"; what does "prognostic" mean? You should call out the specific variable name.

Line 38: remote sensing instruments aboard in satellite.

Line 41: can you quantify the pollution level by giving a range of PSI?

Line 47: remove "(Oozeer et al., 2016)".

Line 48: correlation coefficient of what and what?

Line 49: remove one "(" before "Morrison".

Line 57-59: Clarify the meaning of this sentence "Among . . .".

Line 60-62: please refer this link http://tim.thorpeallen.net/Courses/Reference/Citations.html, or consult with your ACP editor about how to cite a reference.

Line 67: what is PREP-CHEM-SRC? You might not mention it here but give more details in your method part.

Line 73: change "response" to "sensitivity".

Line 74: clarify your purpose here?

Line 87: Should be something like ". . .adapted from the study by Lin et al. (2009)". Refer to the suggestion for Line 60.

Line 92: What is your reference? (It is a good practice when preparing a manuscript, but not in a submitted version).

Line 92: Cloud microphysics?

Line 93-94: Clarify this sentence.

Line 105: What kinds of aerosol species are included in your model?

Line 121" Remove ";"

Line 122: Remove one "(".

Line 132: What is the version of EDGAR?

Line 139: Reference?

Line 145: Based on what you described here, my understand is that your simulation was actually carried freely without nudging to reanalysis for 15-day period using the initial conditions from reanalysis at 00UTC on 15 June 2013. Right? If that is the case, you may not need to mention the restart issue. It causes confusion with the way how the daily forecast performs.

Line 174: Figure 1 – top right and bottom right (not left)?

Line 188: Should be "were used to evaluate surface parameters from simulations…" (The logic should be in a way that simulations are evaluated by observations).

Line 218: The 20kmX panel is not necessary to be shown in the Figure 3. Is it the same as 20km? or you need to mention it.

Line 233: What is 0.017?

Line 292: In Figure 4, what is the right column? You should introduce this experiment at the earlier stage instead of in line 313.

Line 318: How does the factor of 1.3 work? This issue goes back again to the comment on the necessity to introduce the experiments in your method section proceeding the result.

Line 327: A little bit confusion here: to my understand, easterly means that the wind blows from the east to the west. Do you mean eastward instead (blow from the west to the east in your case)? And please double check the description of your wind direction in the line 333 and throughout your manuscript.

Line 322: In Figure 8, why do you use MERRA-2 AOD (an reanalysis) as a reference? Why you don't use AOD from any satellite observations, say MODIS or VIIRS?

Line 326: Do you have evidence on this statement? In addition, you Figure 4-5 don't include June 19.

Line 358-360: You are comparing PM concentration, instead of PM emission, right? They are quite different variables.

Line 361-362: Based on your figures 6-8. Increasing the emission will help to bring the PM concentration level closer to the observation at Brunei only, but overestimates PM in Singapore and overestimates the AOD entirely. Thus increasing the emission seemingly doesn't help much. Or the factor of 6 is too large?

Line 363: Figure 8 (not 7)?

Line 364-365: clarify this sentence. Again, PM emission is different from PM concentration. They are related but not interchangeable.

Line 368: add references here.

Line 610: Figure 6, why the observation in the third panel is different from that in other two panels? Here are suggestions to improve this figure: 1) Combine three panels into one panel by putting 4 time series in one panel; 2) Better to connect the dots into lines. It is difficult to track dots; 3) enlarge the labels in x and y axis. These suggests also apply to the Figure 7. In addition, you talked about Brunei first and then Singapore in the preceding text, then you should follow this same sequence throughout the manuscript. So please switch Figure 6 and 7.
* * *